# A histone H3K9M mutation traps histone methyltransferase Clr4 to prevent heterochromatin spreading

Chun-Min Shan[1†], Jiyong Wang[1†], Ke Xu[1†], Huijie Chen[2], Jia-Xing Yue[3], Stuart Andrews[2], James J Moresco[4], John R Yates III[4], Peter L Nagy[2], Liang Tong[1*], Songtao Jia[1*]

[1]Department of Biological Sciences, Columbia University, New York, United States; [2]Department of Pathology, Columbia University, New York, United States; [3]Institute for Research on Cancer and Aging, Nice (IRCAN), CNRS UMR 7284, INSERM U1081, Nice, France; [4]Department of Chemical Physiology, Scripps Research Institute, La Jolla, United States

**Abstract** Histone lysine-to-methionine (K-to-M) mutations are associated with multiple cancers, and they function in a dominant fashion to block the methylation of corresponding lysines on wild type histones. However, their mechanisms of function are controversial. Here we show that in fission yeast, introducing the K9M mutation into one of the three histone H3 genes dominantly blocks H3K9 methylation on wild type H3 across the genome. In addition, H3K9M enhances the interaction of histone H3 tail with the H3K9 methyltransferase Clr4 in a SAM (S-adenosyl-methionine)-dependent manner, and Clr4 is trapped at nucleation sites to prevent its spreading and the formation of large heterochromatin domains. We further determined the crystal structure of an H3K9M peptide in complex with human H3K9 methyltransferase G9a and SAM, which reveales that the methionine side chain had enhanced van der Waals interactions with G9a. Therefore, our results provide a detailed mechanism by which H3K9M regulates H3K9 methylation.

*For correspondence: ltong@columbia.edu (LT); jia@biology.columbia.edu (SJ)

[†]These authors contributed equally to this work

**Competing interests:** The authors declare that no competing interests exist.

## Introduction

Residue-specific posttranslational modifications of histones play essential roles in every aspect of DNA metabolism such as transcription, DNA replication, recombination, and DNA damage repair (*Lee et al., 2010*). Among these modifications, lysine methylation is the most intriguing due to its remarkable specificity (*Greer and Shi, 2012*). For example, methylation of histone H3 lysine 9 (H3K9me) is critical for the assembly of constitutive heterochromatin at repetitive DNA elements, and H3K27me is essential for the formation of facultative heterochromatin at developmentally regulated genes. The methylation is controlled by specific histone methyltransferases and histone demethylases, mutations of which have been associated with many human diseases (*Greer and Shi, 2012*; *Herz et al., 2013*). In comparison, histone genes are not prime suspects for disease-associated mutations, as their high copy number masks the effects of any recessive changes. However, recent high throughput sequencing analyses identified a high incidence of somatic mutations of critical histone lysines into methionine (K-to-M) in multiple cancers (*Schwartzentruber et al., 2012*; *Sturm et al., 2012*; *Wu et al., 2012*; *Behjati et al., 2013*; *Shah et al., 2014*). The most prominent examples include the association of a K27M mutation in H3.3 (a variant form of H3) with pediatric high-grade glioblastoma, a highly aggressive brain tumor, and the association of a K36M mutation in H3.3 with chondroblastoma, a tumor of the bone. The high frequency of such mutations indicates that they are driver mutations of tumorigenesis.

**eLife digest** Cells switch their genes on or off in order to respond to changes in their environment. A group of proteins called histones are partly responsible for regulating gene activity. Like all proteins, histones are made from smaller building blocks called amino acids. Enzymes can chemically modify specific amino acids in histone proteins, which allows the histones to switch nearby genes on or off. One of these modifications is called methylation, and the methylation of specific "lysine" amino acids in histone proteins regulates gene activity in different ways.

Previous research has shown that, in certain types of cancer cells, lysines that can be methylated are often replaced with another amino acid, a methionine. These substitutions stop the histones from correctly controlling the activity of nearby genes because methionine cannot be methylated like lysine. Additionally, even if only a small number of histones have methionine in place of lysine, this change can have a widespread effect because the few histones with the methionine can block other histones from being methylated too. However, previous studies did not provide a clear mechanism for why this is the case.

In the fission yeast *Schizosaccharomyces pombe* an enzyme called Clr4 methylates a histone protein at a lysine named H3K9. Now, Shan, Wang, Xu et al. show that substituting this lysine with a methionine (referred to as H3K9M) stops the widespread methylation of histones by trapping the Clr4 enzyme. Specifically, Clr4 becomes stuck to the H3K9M histones, and is therefore unable to modify any other histones. Shan et al. went on to carry out a more detailed study of the structure of H3K9M attached to another enzyme called G9a. This enzyme is found in human cells and is similarly inhibited by H3K9M. This investigation identified additional chemical interactions that explain why Clr4 and G9a become trapped by the H3K9M histone but not by normal histones.

Future studies are needed to explore whether other altered histones are able to trap enzymes in the way that H3K9M traps Clr4 and G9a. In addition, this work could eventually lead to new cancer therapies.

---

The role of histone K-to-M mutations in cancer is under intense investigation, but the mechanism is not well understood and controversial. For example, cells containing H3.3K27M show reduced H3K27 trimethylation (H3K27me3) levels on wild type histones, and an H3K27M peptide inhibits polycomb repressive complex 2 (PRC2)-mediated H3K27 methylation in vitro, suggesting that these mutations function as inhibitors of H3K27 methyltransferases (*Bender et al., 2013*; *Chan et al., 2013a*; *Lewis et al., 2013*; *Venneti et al., 2013*). However, ChIP-seq analyses of cancer cells containing H3.3K27M show that in addition to a global reduction of H3K27me3, a large number of ectopic H3K27me3 peaks are also detected, arguing against a simple inhibitory model (*Bender et al., 2013*; *Chan et al., 2013a*).

Other histone K-to-M mutations also dominantly block the methylation of their corresponding lysine residues (*Chan et al., 2013b*; *Lewis et al., 2013*; *Herz et al., 2014*; *Lu et al., 2016*; *Fang et al., 2016*), suggesting a similar mechanism of function. Interestingly, mass spectrometry analyses of H3K9M-containing nucleosomes purified from human cells show increased association of an H3K9 demethylase KDM3B, suggesting that recruitment of a demethylase contributes to the effects of H3K9M on H3K9 methylation (*Herz et al., 2014*).

One of the difficulties in reconciling these results is the complication of the mammalian system. First, there are multiple copies of genes encoding histone H3 and H3.3 (*Maze et al., 2014*), but it is not clear whether each gene is equivalent in its expression profile. Second, there are multiple methyltransferases that modify each lysine (*Greer and Shi, 2012*), and they are differentially inhibited by K-to-M mutations (*Fang et al., 2016*).

Another difficulty in studying the mechanism of histone K-to-M mutations is the paucity of structural data that explains how the methionine substitution functions to inhibit histone methyltransferases. There are two recent structures of an H3K27M peptide in association with the H3K27 methyltransferase PRC2, which give conflicting interaction details. The structure of *Chaetomium thermophilum* PRC2 reveals that an arginine residue (R26) adjacent to H3K27M occupies the expected lysine-binding channel (*Jiao and Liu, 2015*), whereas the structure of human PRC2

indicates that the methionine side chain occupies this channel (*Justin et al., 2016*). There is currently no structure that examines the interaction between other K-to-M mutations and their corresponding histone methyltransferases.

The fission yeast *Schizosaccharomyces pombe* shares highly conserved chromatin modification pathways with mammals, but has key advantages such as relatively facile genetics and single representatives of most key families of mammalian chromatin-modifying factors (*Wood et al., 2002*). For example, a single histone H3K9 methyltransferase Clr4 is critical for H3K9me across the entire genome and regulates the assembly of heterochromatin (*Rea et al., 2000*; *Nakayama et al., 2001*; *Cam et al., 2005*). Moreover, there are only three copies of histone H3 genes with identical protein sequences (*Mellone et al., 2003*). The simplicity of the fission yeast system allows us to demonstrate that H3K9M blocks H3K9 methylation through trapping of Clr4 at heterochromatin nucleation centers to prevent its spreading. In addition, H3K9M interacts directly with Clr4 and the interaction is dramatically enhanced by S-adenosylmethionine (SAM), the methyl donor for histone methyltransferases. Moreover, we determined a high-resolution crystal structure of H3K9M in complex with an H3K9 methyltransferase G9a and SAM, which provides a detailed mechanism for the role of methionine in trapping histone methyltransferases. These results provide clear mechanistic insights of histone H3K9M in regulating H3K9me levels.

## Results

### H3K9M dominantly blocks H3K9 methylation and heterochromatin assembly

Fission yeast contains three histone H3 genes: *hht1⁺*, *hht2⁺*, and *hht3⁺*, which produce identical proteins (*Mellone et al., 2003*). We generated Flag-tagged versions of each of these genes at their endogenous chromosomal loci. Western blot analyses showed that the three proteins were expressed at similar levels (*Figure 1—figure supplement 1*). We then introduced the K9M mutation into each histone H3 gene at its endogenous chromosome locus. The three mutant histone H3s were also expressed at similar levels (*Figure 1—figure supplement 1*).

In fission yeast, heterochromatin is mainly present at pericentric region, subtelomeres, and the silent mating-type region, which all contain a similar repetitive DNA element (*Grewal and Jia, 2007*). These regions contain high levels of histone H3K9 methylation and the transcription of the underlying repeats is repressed. Reporter genes inserted within these repeats, such as *otr::ura4⁺* inserted at the *dh* repeat of pericentric region and *Kint2::ura4⁺* inserted at the *cenH* repeat of the mating-type region (*Figure 1A*) (*Allshire et al., 1995*; *Grewal and Klar, 1997*), are silenced, resulting in cells that only grow weakly on medium without uracil. These cells grow well on medium containing 5-fluoroorotic acid (FOA), which is converted to a toxic form (5-flurouracil) in cells expressing *ura4⁺* (*Figure 1B*, and *Figure 1—figure supplement 2*). Loss of heterochromatin, such as in *clr4Δ*, results in the expression of these reporters and robust cell growth on medium without uracil and decreased growth on FOA-containing medium. Introducing a K9M mutation into any H3 gene resulted in silencing defects similar to that of *clr4Δ*, even though there were two other wild type histone H3 genes present (*Figure 1B*, and *Figure 1—figure supplement 2*). Therefore, we used only *hht3-K9M* for all subsequent analyses. In contrast, an *hht3-K9R* mutation (which is expected to abolish methylation of lysine 9 only on Hht3) had little effect on gene silencing (*Figure 1—figure supplement 2*), suggesting that the effect of H3K9M is specific.

ChIP analyses showed that H3K9me3 levels were abolished and H3K9me2 levels were significantly reduced at the heterochromatic *dh* and *cenH* repeats (*Figure 1B*). Moreover, transcripts of these repeats were dramatically increased (*Figure 1B*). Further ChIP-seq analyses showed that H3K9me3 was reduced to background levels across the entire genome, similar to *clr4Δ* (*Figure 1C* and *Figure 1—figure supplement 3*). Unlike the presence of additional H3K27me3 peaks in H3.3K27M containing mammalian cells (*Bender et al., 2013*; *Chan et al., 2013a*), we did not detect any additional peaks of H3K9me3 in *hht3-K9M* cells. Therefore, H3K9M functions dominant-negatively to regulate H3K9 methylation across the genome.

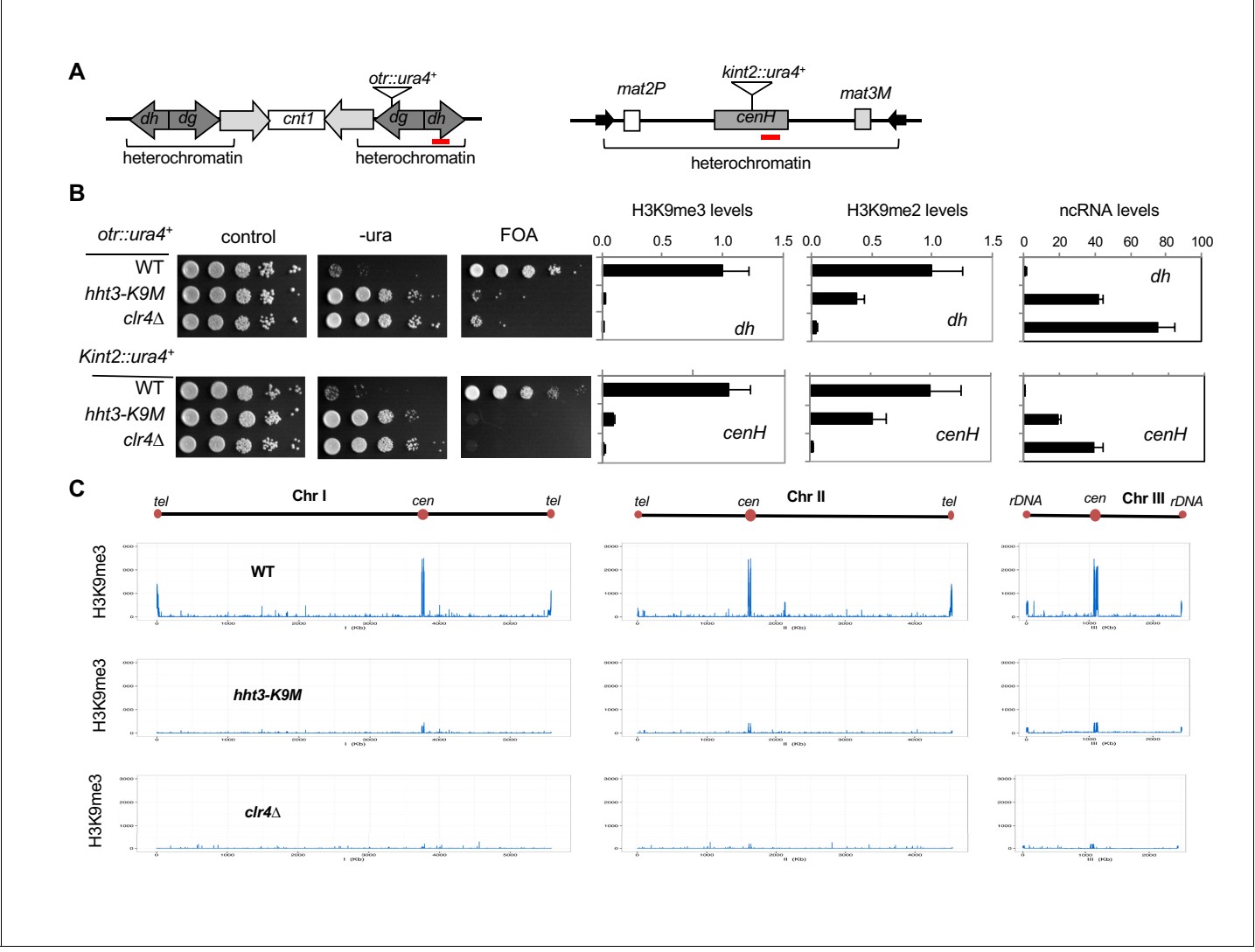

**Figure 1.** H3K9M affects H3K9me and heterochromatin assembly. (**A**) Schematic diagrams of reporter genes used. Red bars indicate primers used for ChIP-qPCR and qRT-PCR analyses. (**B**) Left, ten-fold serial dilution analyses of indicated yeast strains grown on indicated media to measure the expression of *otr::ura4⁺*. Right, ChIP analyses of H3K9me3 and H3K9me2 levels at heterochromatin-associated repeats, normalized to *act1⁺*, and qRT-PCR analysis of repeat transcript levels, normalized to *act1⁺*. Data presented is mean ± s.d. of four technical repeats. (**C**) ChIP-seq analysis of H3K9me3 levels across the genome. Data presented is average of two sequencing runs.

The following figure supplements are available for figure 1:

**Figure supplement 1.** Western blot analyses of histone H3 and Tubulin levels.

**Figure supplement 2.** Ten-fold serial dilution analyses of indicated yeast strains grown on indicated media to measure the expression of *otr::ura4⁺*.

**Figure supplement 3.** ChIP-seq analyses of H3K9me3 levels at centromeres and telomeres.

## H3K9M blocks the enzymatic activity of Clr4

It has been shown that an H3K27M peptide inhibits the enzymatic activity of the H3K27 methyltransferase PRC2 in vitro (*Lewis et al., 2013*). To examine whether H3K9M also similarly inhibits the enzymatic activity of Clr4, we performed in vitro histone methyltransferase assays using a recombinant Clr4 SET domain (amino acids 190–490), recombinant mono-nucleosomes, and $^3$H labelled S-adenosylmethionine (SAM). The incorporation of radioactively labelled methyl group into histone H3 was

significantly reduced in the presence of an H3K9M peptide, suggesting that H3K9M directly inhibits the enzymatic activity of Clr4 in vitro (*Figure 2A*).

To test the inhibitory effects of H3K9M in vivo, we used an *ade6+* reporter that is adjacent to 3 copies of Gal4-binding sites (*3xgbs-ade6+*). In wild type cells, the reporter gene is expressed, resulting in the formation of white colonies on low adenine medium (YE). In contrast, artificially targeting

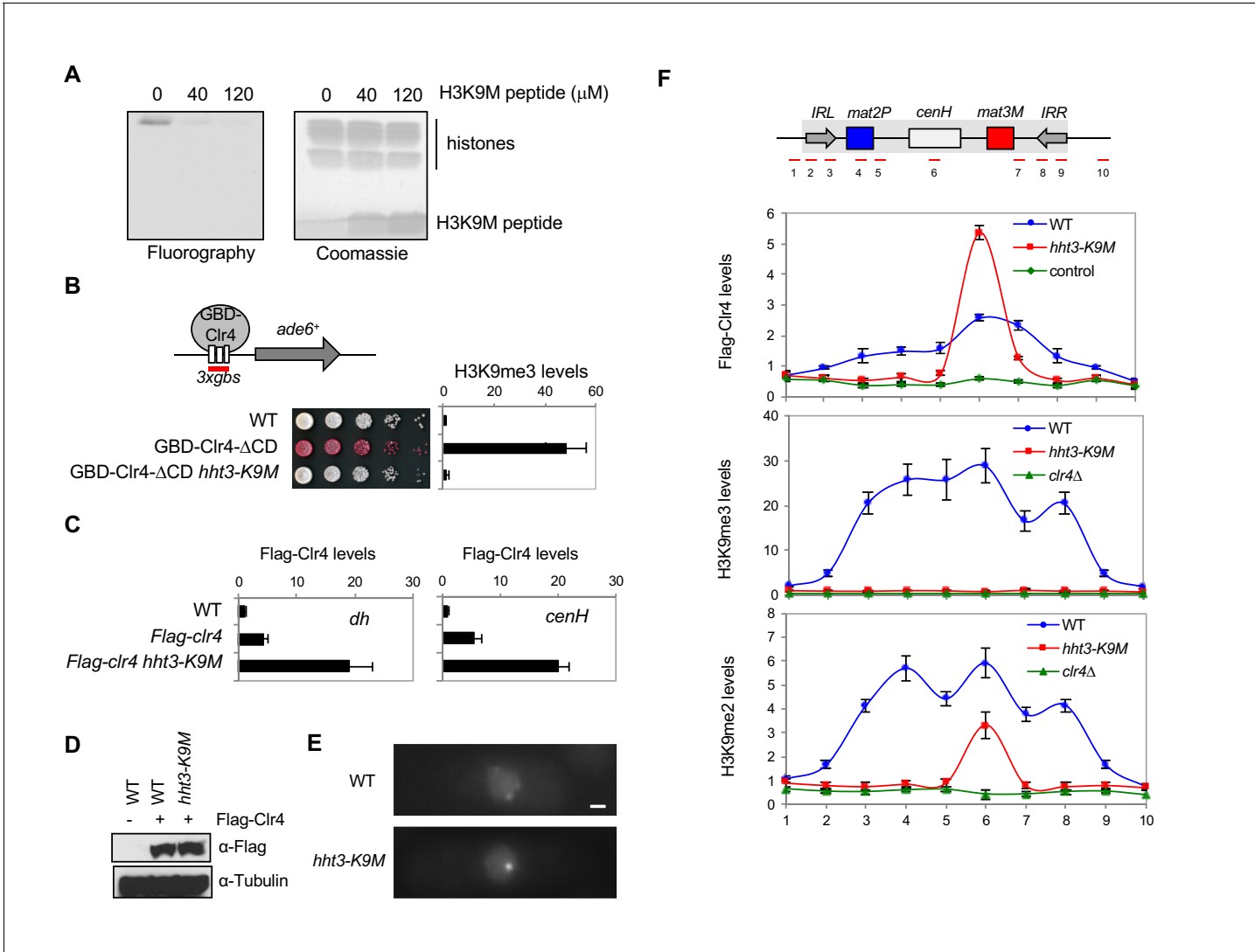

**Figure 2.** H3K9M traps Clr4 at nucleation centers to prevent heterochromatin spreading. (**A**) In vitro histone methyltransferase assay with recombinant Clr4 SET domain and recombinant nucleosomes. (**B**) Top, a schematic diagram of the *3xgbs-ade6+* reporter. Left, ten-fold serial dilution analyses of indicated yeast strains grown on low adenine media (YE) to measure the expression of *3xgbs-ade6+*. Right, ChIP analyses of H3K9me3 levels at *3xgbs*, normalized to *act1+*. Data presented is mean ± s.d. of four technical replicates. (**C**) ChIP analysis of Clr4 levels at heterochromatin associated repeats, normalized to *act1+*. Data presented is mean ± s.d. of four technical replicates. (**D**) Western blot analysis of Flag-Clr4 protein levels. (**E**) Live cell imaging of cells expressing GFP-Clr4. Scale bar is 1 μm. (**F**) Top, a schematic diagram of the mating type region, with shaded area representing heterochromatin. The PCR fragments used for qPCR analyses are labeled. Bottom, ChIP analysis of Flag-Clr4, H3K9me3, and H3K9me2 levels across the mating type region, normalized to *act1+*. Data presented is mean ± s.d. of four technical replicates.

The following figure supplements are available for figure 2:

**Figure supplement 1.** Yeast strains were grown on low adenine media (YE) to measure the expression of *3xgbs-ade6+*.

**Figure supplement 2.** H3K9M affects mating-type switching.

Clr4 through a Gal4 DNA binding domain silences the reporter, resulting in red colonies (*Kagansky et al., 2009*). In *hht3-K9M* cells, the reporter gene was not silenced and no H3K9me3 was detected (*Figure 2B* and *Figure 2—figure supplement 1*), suggesting that H3K9M inhibits the enzymatic activity of Clr4 in vivo.

## H3K9M traps Clr4 at heterochromatin nucleation centers and prevents Clr4 spreading

Surprisingly, ChIP analyses showed that Clr4 protein levels were much higher at *dh* and *cenH* repeats in *hht3-K9M* cells (*Figure 2C*), even though overall Clr4 protein levels were unchanged (*Figure 2D*). Consistently, live-cell imaging analyses showed stronger GFP-Clr4 foci in *hht3-K9M* cells (*Figure 2E*). These results suggest that *hht3-K9M* does not affect the targeting of Clr4 to chromatin, but instead traps Clr4 at heterochromatic locations.

Heterochromatin formation is divided into two distinct steps: initiation and spreading (*Wang et al., 2014*). First, Clr4 is recruited to nucleation centers to initiate H3K9me. H3K9me recruits additional Clr4, through the chromo domain of Clr4 and Swi6 proteins, leading to methylation of adjacent nucleosomes (*Hall et al., 2002*; *Zhang et al., 2008a*; *Al-Sady et al., 2013*). The repetition of such cycles promotes heterochromatin spreading across large chromosome domains. Both *dh* and *cenH*, which showed higher levels of Clr4, are heterochromatin nucleation centers (*Hall et al., 2002*). To investigate the effects of H3K9M on heterochromatin spreading, we measured Clr4, H3K9me3, and H3K9me2 levels at the silent mating-type region, which has been extensively used to examine heterochromatin initiation and spreading (*Hall et al., 2002*). At this region, *cenH* recruits Clr4, which then spreads into a 20 kb domain that are marked by two inverted repeats (*IRL* and *IRR*) (*Figure 2F*). We found that in *hht3-K9M* cells, Clr4 was highly enriched at *cenH*, but not at the surrounding regions (*Figure 2F*). Furthermore, H3K9me3 was completely abolished across the entire locus, whereas low levels of H3K9me2 were restricted to *cenH* (*Figure 2F*). Therefore, H3K9M traps Clr4 at nucleation centers and prevents its spreading into neighboring regions.

Heterochromatin at the silent mating-type locus is essential for the correct choice of donors during mating-type switching, when heterochromatin-embedded *mat2P* or *mat3M* are used as a donor to replace the DNA sequence at *mat1* (*Jia et al., 2004*) (*Figure 2—figure supplement 2A*). Loss of heterochromatin leads to the predominant use of *mat3M* as a donor, the accumulation of the *M* mating type within a switching competent population, and reduced mating efficiency (indicated by lightly iodine-stained colonies on medium that induces mating and meiosis) (*Jia et al., 2004*) (*Figure 2—figure supplement 2B and C*). As expected, *hht3-K9M* cells were predominantly of the *M* mating type and formed lightly iodine-stained colonies similar to *clr4Δ* cells (*Figure 2—figure supplement 2B and C*).

## H3K9M enhances binding of Clr4 SET domain to histone H3 tail in a SAM-dependent manner

One possible reason for the increased levels of Clr4 at heterochromatin nucleation centers is that Clr4 has a higher affinity for H3K9M. However, the fact that H3K9M-containing nucleosomes outside of heterochromatin failed to trap Clr4 suggests that the interaction between Clr4 and H3K9M is regulated. Clr4 contains a catalytic SET domain as well as a chromo domain that interacts with H3K9me. We found that the chromo domain of Clr4 (1–190) interacted with an H3K9me3 peptide and to a lesser extent with an H3K9me2 peptide, but showed no interaction with the H3K9M peptide (*Figure 3A*). In addition, none of the other H3K9me-interacting chromo domains in fission yeast, including Swi6, Chp2, and Chp1 (*Sadaie et al., 2008*; *Schalch et al., 2009*), interacted with the H3K9M peptide (*Figure 3—figure supplement 1*). Interestingly, the SET domain of Clr4 (190–490) showed stronger interaction with an H3K9M peptide than a wild type histone H3 tail peptide in a histone methyltransferase buffer (50 mM Tris, pH 8.0, 1 mM EDTA, 50 mM NaCl, 1 mM DTT) supplemented with 100 μM SAM (*Figure 3B*). In addition, such interaction was significantly reduced in the absence of SAM (*Figure 3B*). The binding is not covalent as Clr4 was removed under stringent washing conditions (data not shown). Moreover, the binding is significantly reduced when S-adenosylhomocysteine (SAH) was used in place of SAM (*Figure 3C*). Furthermore, the SET domain of human H3K9 methyltransferase G9a, which is inhibited by H3K9M in vitro (*Lewis et al., 2013*), also showed

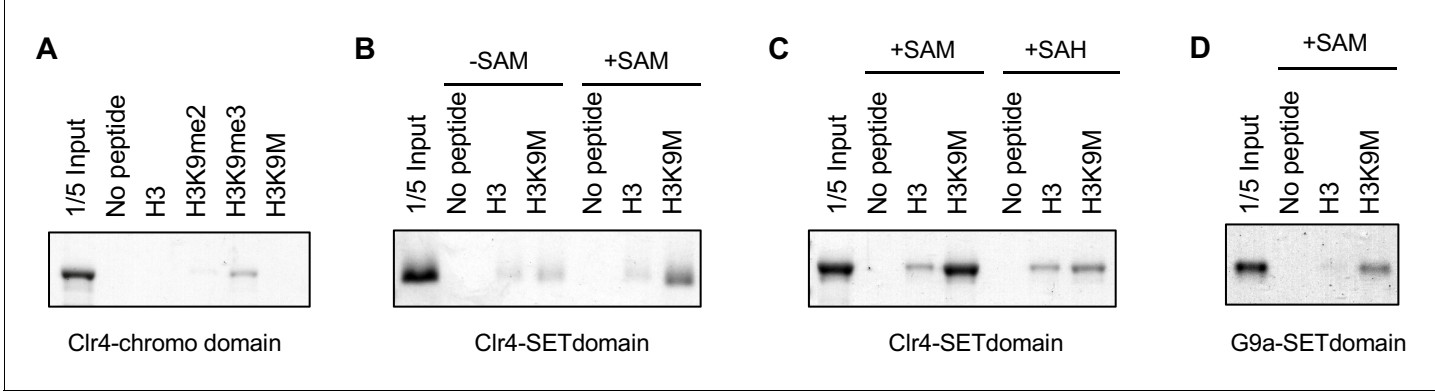

**Figure 3.** H3K9M interacts directly with H3K9 methyltransferases. (A) Binding assays with recombinant Clr4 chromo domain and histone peptides in PBS buffer. (B, C) Binding assays with recombinant Clr4 SET domain and histone peptides in histone methyltransferase buffer supplemented with SAM or SAH. (D) Binding assays with recombinant G9a SET domain and histone peptides in histone methyltransferase buffer supplemented with SAM.
The following figure supplements are available for figure 3:

**Figure supplement 1.** Binding assays with recombinant GST-tagged chromo domains and histone peptides in PBS buffer.
**Figure supplement 2.** H3K9M functions through the SET domain of Clr4.

stronger binding to the H3K9M peptide than the wild type H3 tail peptide in the presence of SAM (*Figure 3D*), suggesting a similar binding mechanism.

The binding data suggests that H3K9M exerts its effects through the SET domain of Clr4. To rule out the contribution of the chromo domain of Clr4 to the trapping of Clr4 by H3K9M in vivo, we generated a W31G mutation within the chromo domain, which abolishes its interaction with H3K9me3, but does not affect the enzymatic activity of Clr4 in vitro (*Nakayama et al., 2001*; *Zhang et al., 2008a*). Similar to previous observations, introducing the W31G mutation at the endogenous *clr4⁺* locus resulted in a severe reduction of H3K9me3, but had little effects on H3K9me2 at *cenH*, the heterochromatin nucleation center of the silent mating-type region (*Figure 3—figure supplement 2A*) (*Al-Sady et al., 2013*). The localization of Clr4 to *cenH* was severely reduced in *clr4-W31G* cells, due to the dependence of H3K9me3 for the stable association of Clr4 with heterochromatin (*Figure 3—figure supplement 2B*) (*Zhang et al., 2008a*). Consistent with the idea that H3K9M exerts its effects through the Clr4 SET domain, Clr4-W31G showed enhanced localization at *cenH* in *hht3-K9M* cells (*Figure 3—figure supplement 2C*).

## Structure of H3K9M in complex with H3K9 methyltransferase G9a and SAM

To understand the molecular details for the interactions between H3K9M and its methyltransferases, we determined a 1.7 Å resolution crystal structure of G9a SET domain in complex with an H3K9M peptide (ARTKQTARMSTGGKA) and SAM (*Supplementary file 1*). Clear electron density was observed for residues 3–12 of the peptide based on the crystallographic analysis (*Figure 4A*). We did not include additional SAM during purification and crystallization, but a SAM molecule was observed in the electron density (*Figure 4B*), which was likely copurified with the G9a SET domain from *E. coli* lysates.

Compared with the structure of another H3K9 methyltransferase G9a-like protein 1 (GLP1) in complex with a dimethylated H3K9 peptide and SAH (*Wu et al., 2010*), the binding mode of the H3K9M peptide to G9a is essentially identical (*Figure 4C and D*). The methionine residue of H3K9M occupies the same binding pocket as the dimethylated lysine residue (*Figure 4D*). Five aromatic residues, Tyr1067, Tyr1087, Phe1152, Tyr1154 and Phe1158 surround the methionine side chain. While the dimethylated Lys side chain assumes a fully extended conformation, there is a sharp bend at the sulfur atom of methionine, such that its end methyl group is in closer contact with Tyr1067 and the sulfur is in closer contact with Phe1152 and Tyr1154. The methionine side chain does not extend as

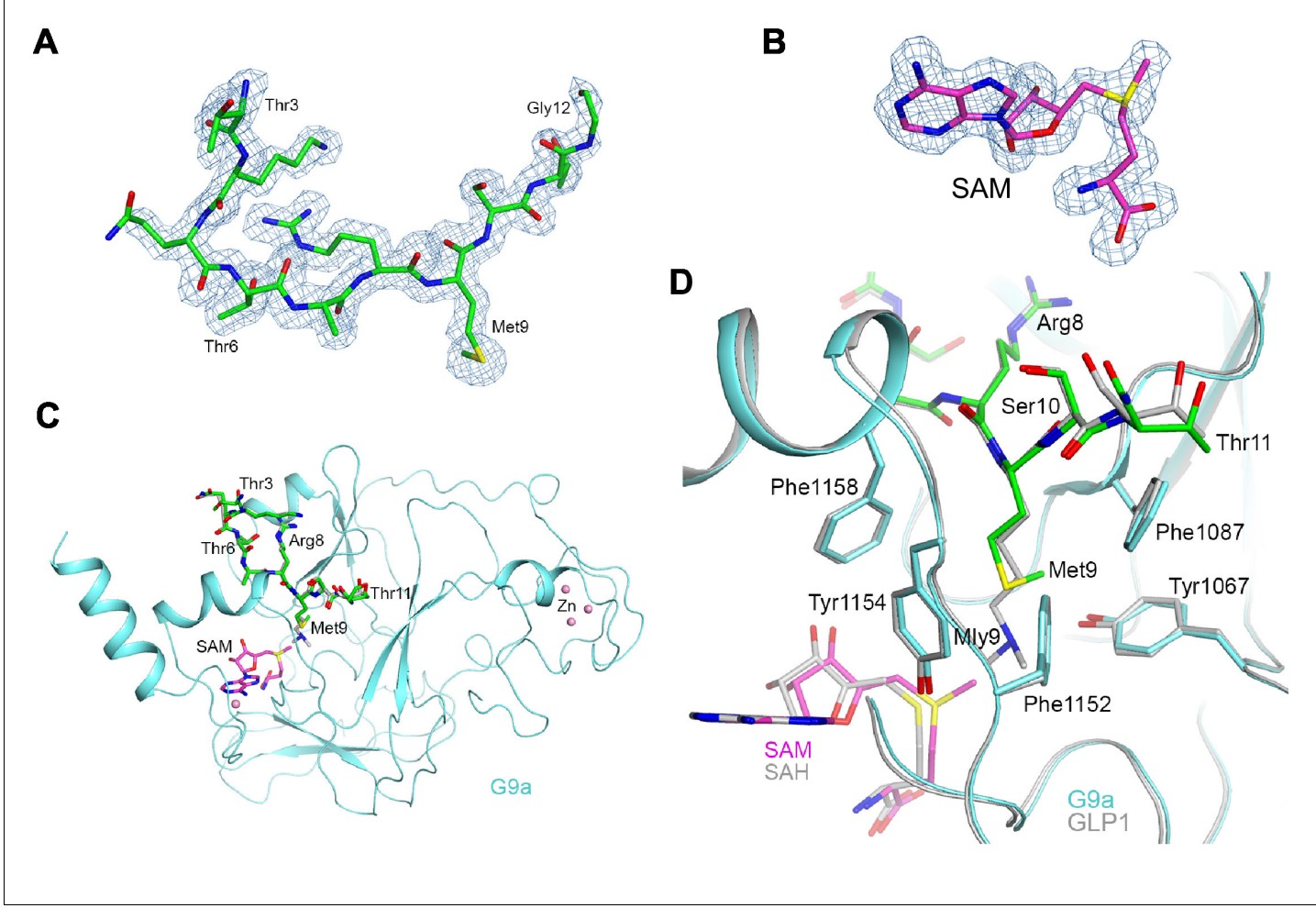

**Figure 4.** Structure of the H3K9M-G9a-SAM complex. (**A**) Omit $F_o$–$F_c$ electron density at 1.7 Å resolution for the H3K9M peptide, contoured at 3σ. (**B**) Omit $F_o$–$F_c$ electron density at 1.7 Å resolution for SAM, contoured at 3σ. (**C**) Overall structure of G9a (cyan) in complex with the H3K9M peptide (green, stick models) and SAM (magenta). Several zinc ions bound to the protein are shown as pink spheres. The bound position of dimethylated H3K9 peptide to GLP1 is also shown (gray). (**D**) Detailed interactions between G9a (cyan) and the H3K9M peptide (green) near Met9. The structure of GLP1 in complex with the dimethylated H3K9 (Mly9) peptide and SAH is also shown (gray).

The following figure supplements are available for figure 4:

**Figure supplement 1.** Structure of the H3K9Nle-G9a-SAM complex.

**Figure supplement 2.** Sequence alignment of SUV39 family H3K9 histone methyltransferases.

**Figure supplement 3.** Characterization of the Clr4-Y451N mutant.

deeply into the pocket, and the SAM compound at the bottom of the pocket moves closer to the side chain, by ~0.8 Å compared to the position of SAH in GLP1. SAM does not make direct contributions to the interactions with the methionine side chain, suggesting that its role is to stabilize the structure of the protein. Interestingly, the methyl group of SAM is 3.4 Å away from the side chain of Tyr1154, which is part of the conformational change to stabilize the peptide-binding pocket. In contrast, the G9a-SAH structure does not have this interaction (*Wu et al., 2010*).

It has been demonstrated that the hydrophobicity of the side chain is critical for H3K27M to inhibit PRC2, and changing methionine to norleucine produces a more potent inhibitor (*Lewis et al., 2013*; *Brown et al., 2014*). To further understand the contribution of the aliphatic side chain to the

interaction between H3K9M and G9a, we performed binding assays with an H3K9-norleucine (H3K9Nle) peptide. We found that H3K9Nle also enhanced binding of the H3 tail peptide to G9a similar to H3K9M (data not shown). We also solved a 1.9Å structure of G9a in complex with an H3K9Nle peptide and SAM (*Figure 4—figure supplement 1*). The structure showed that the norleucine side chain adopts a similar orientation as the methionine side chain, with a bend of the end methyl group (*Figure 4—figure supplement 1*).

To characterize the importance of residues in the binding site for the interactions with the H3K9M peptide, we mutated Tyr451 of Clr4, which is equivalent to Tyr1154 of G9a, to asparagine (*Figure 4—figure supplement 2*). Consistent with our expectations, the binding of Clr4-SET-Y451N to the H3K9M peptide was significantly reduced (*Figure 4—figure supplement 3A*). We then introduced the Y451N mutation into the endogenous *clr4⁺* locus. The mutant protein was expressed at similar levels as that of wild-type Clr4 and mass spectrometry analysis showed efficient association with other components of the Clr4 complex (*Hong et al., 2005*; *Horn et al., 2005*; *Jia et al., 2005*) (*Figure 4—figure supplement 3B and C*). The localization of Clr4-Y451N to pericentric *dh* repeats was completely abolished, and no trapping of Clr4 was observed in *hht3-K9M* cells (data not shown). The complete loss of localization of Clr4-Y451N to *dh* repeats is due to the positive feedback mechanism between H3K9 methylation and Clr4 localization (*Zhang et al., 2008a*), as the Y451N mutation cannot methylate histones in vitro and in vivo (*Figure 4—figure supplement 3D and E*). Therefore, although the Y451N mutant demonstrates the critical role of this residue in mediating interaction with H3K9M in vitro, it cannot distinguish in vivo the contribution of its being catalytically inactive or inability to bind H3K9M. We have also mutated two other residues of Clr4 that are expected to interact with H3K9M, Y357N and F383N. However, both of them behaved similarly as Y451N, with complete loss of enzymatic activity in vitro and in vivo, but associated efficiently with other components of the Clr4 complex (data not shown).

## Alleviating Clr4 trapping overcomes the inhibitory effects of H3K9M

Given that the aromatic residues interacting with the methionine side chain are also required for interaction with the normal lysine substrate, it is not surprising that mutating these residues affected Clr4 enzymatic activity. We therefore explored other ways to relieve trapping of Clr4 by H3K9M. The enzymatic activity of Clr4 in vitro is influenced by modifications of residues adjacent to H3K9 such as phosphorylation of serine 10 (*Nakayama et al., 2001*). We suspected that phosphorylated S10 might interfere with the binding of Clr4 with H3K9M peptide. Indeed, in vitro binding of Clr4 SET domain was significantly reduced when an H3K9M peptide containing phosphorylated S10 was used (*Figure 5A*). In addition, the in vitro binding of Clr4 to an H3K9MS10D peptide (to mimic the phosphorylated state of S10) was also dramatically reduced (*Figure 5A*).

To examine whether alleviating the trapping of Clr4 by H3K9M can detoxify H3K9M in vivo, we constructed an *hht3-K9MS10D* transgene to mimic the constitutively phosphorylated S10 in vivo. The mutant histone was expressed at similar levels to *hht3-K9M*, and the mutation had no effects on Clr4 protein levels (*Figure 5B*). Consistent with our hypothesis, Clr4 was localized to pericentric *dh* repeat in *hht3-K9MS10D* cells at levels very similar to those in wild-type cells, demonstrating that trapping by H3K9M was alleviated by the adjacent S10 modification (*Figure 5C*). Moreover, *hht3-K9MS10D* cells showed normal heterochromatin as indicated by silencing of *otr::ura4⁺*, wild type levels of H3K9me3 and H3K9me2 at pericentric repeats, and wild type levels of *dh* RNA transcripts (*Figure 5D*). The S10D mutation only detoxified H3K9M when it was present *in cis* with K9M, as introducing an *hht1-S10D* mutation *in trans* could not rescue silencing defects associated with *hht3-K9M* (*Figure 5—figure supplement 1*).

## The effects of H3K9M also depends on Clr4 protein levels and the relative concentrations of H3K9M to wild-type H3

To further examine the mechanism by which H3K9M regulates H3K9 methylation, we introduced a genomic DNA library into cells containing *hht3-K9M* and *otr::ura4⁺* and looked for clones that conferred resistance to FOA (*Figure 6A*). We identified two clones, and sequencing of these two plasmids showed that one contains the *clr4⁺* gene and the other contains histone H3 gene *hht2⁺* (*Figure 6B* and *Figure 6—figure supplement 1A*). A similar screen with a cDNA library also identified three suppressors that all encode histone H3 genes (*Figure 6B* and *Figure 6—figure*

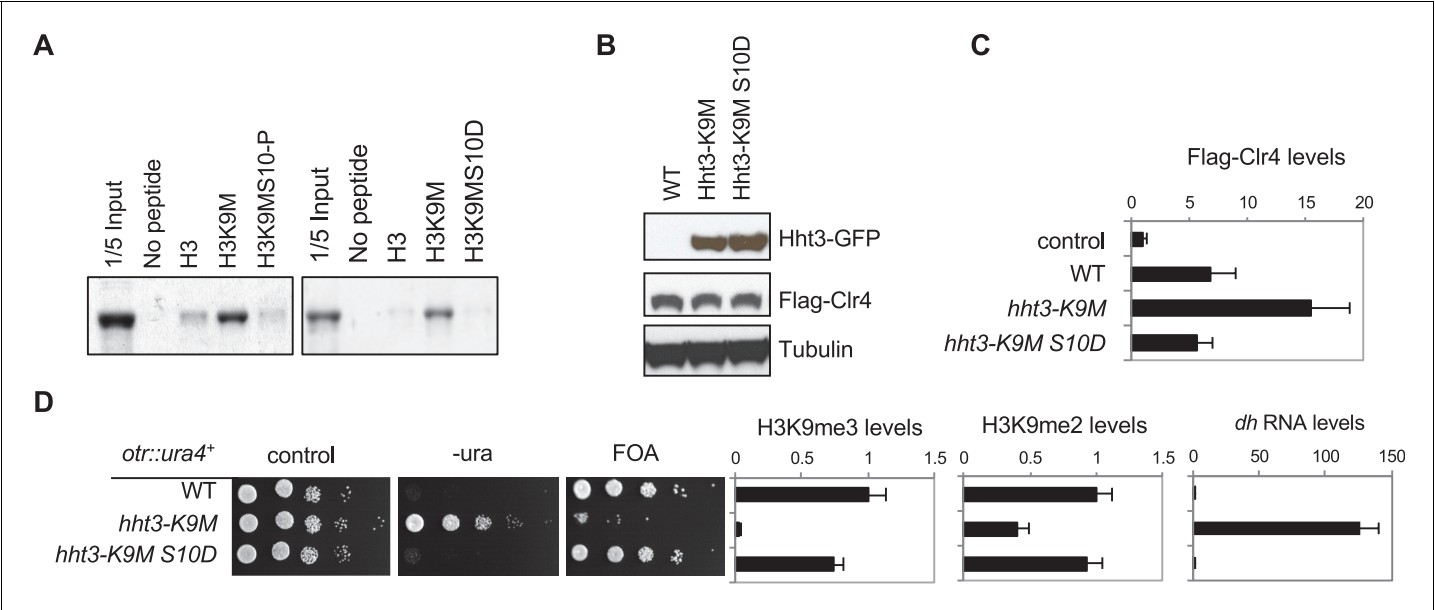

**Figure 5.** Reduced binding between H3K9M and Clr4 alleviates its inhibitory effects. (**A**) Binding assays with recombinant Clr4 SET domain and histone peptides in histone methyltrasnferase buffer containing SAM. (**B**) Western blot analyses of histone H3 and Clr4 levels. (**C**) ChIP analysis of Clr4 levels at heterochromatin associated repeats, normalized to $act1^+$. Data presented is mean ± s.d. of four technical repeats. (**D**) Left, ten-fold serial dilution analyses of indicated yeast strains grown on indicated media to measure the expression of $otr::ura4^+$. Right, ChIP analyses of H3K9me3 and H3K9me2 levels at heterochromatin-associated repeats, normalized to $act1^+$, and qRT-PCR analysis of repeat transcript levels, normalized to $act1^+$. Data presented is mean ± s.d. of four technical repeats.

The following figure supplement is available for figure 5:

**Figure supplement 1.** S10D has to be present *in cis* with K9M to detoxify H3K9M.

*supplement 1A*). These results raise the possibility that the relative dosage of Clr4 and histone H3 is essential for the phenotypes associated with the H3K9M mutation.

To further test the effects of Clr4 dosage on *hht3-K9M*, we replaced the *clr4⁺* gene promoter with an inducible *nmt41* promoter. Induction of *clr4⁺* partially rescued the silencing defects associated with *hht3-K9M* (*Figure 6C*). The effect is Clr4 dosage dependent, as the stronger *nmt1* promoter provided better rescue and the weaker *nmt81* promoter conferred less rescue (*Figure 6— figure supplement 1B*). In contrast, overexpression of Swi6, an HP1 family protein that binds to H3K9me (*Bannister et al., 2001*; *Lachner et al., 2001*; *Nakayama et al., 2001*), could not rescue silencing defects associated with *hht3-K9M* (*Figure 6—figure supplement 1B*).

To investigate the effects of histone dosage on *hht3-K9M*, we constructed diploid cells containing one or two copies of *hht3-K9M*. Diploid cells containing two copies of *hht3-K9M* are similar in mutant histone dosage as haploid cells containing one copy of *hht3-K9M*, and behaved similarly (*Figure 6D*). However, diploid cells containing one copy of *hht3-K9M* resulted in only partial desilencing of *otr::ura4⁺*, with normal H3K9me2 and about a 50% reduction of H3K9me3 (*Figure 6D*). Therefore, H3K9M acts in a dosage-dependent manner to affect H3K9 methylation and heterochromatin assembly.

We also tested the effects of histone demethylation in *hht3-K9M* cells. In fission yeast, genetic studies suggest that the JmjC domain protein Epe1 is an H3K9 demethylase, although it should be noted that no demethylase activity of Epe1 is detected in vitro (*Tsukada et al., 2006*; *Zofall and Grewal 2006*; *Trewick et al., 2007*; *Audergon et al., 2015*; *Ragunathan et al., 2015*). We found that *epe1Δ hht3-K9M* cells partially rescued silencing defects of *otr::ura4⁺*, accompanied by the restoration of H3K9me2 levels and a reduction of pericentric *dh* repeat transcript levels, although H3K9me3 levels were only marginally rescued (*Figure 6E*). Therefore, the balance of heterochromatin promoting and destabilizing forces also contribute to the effects of *hht3-K9M*.

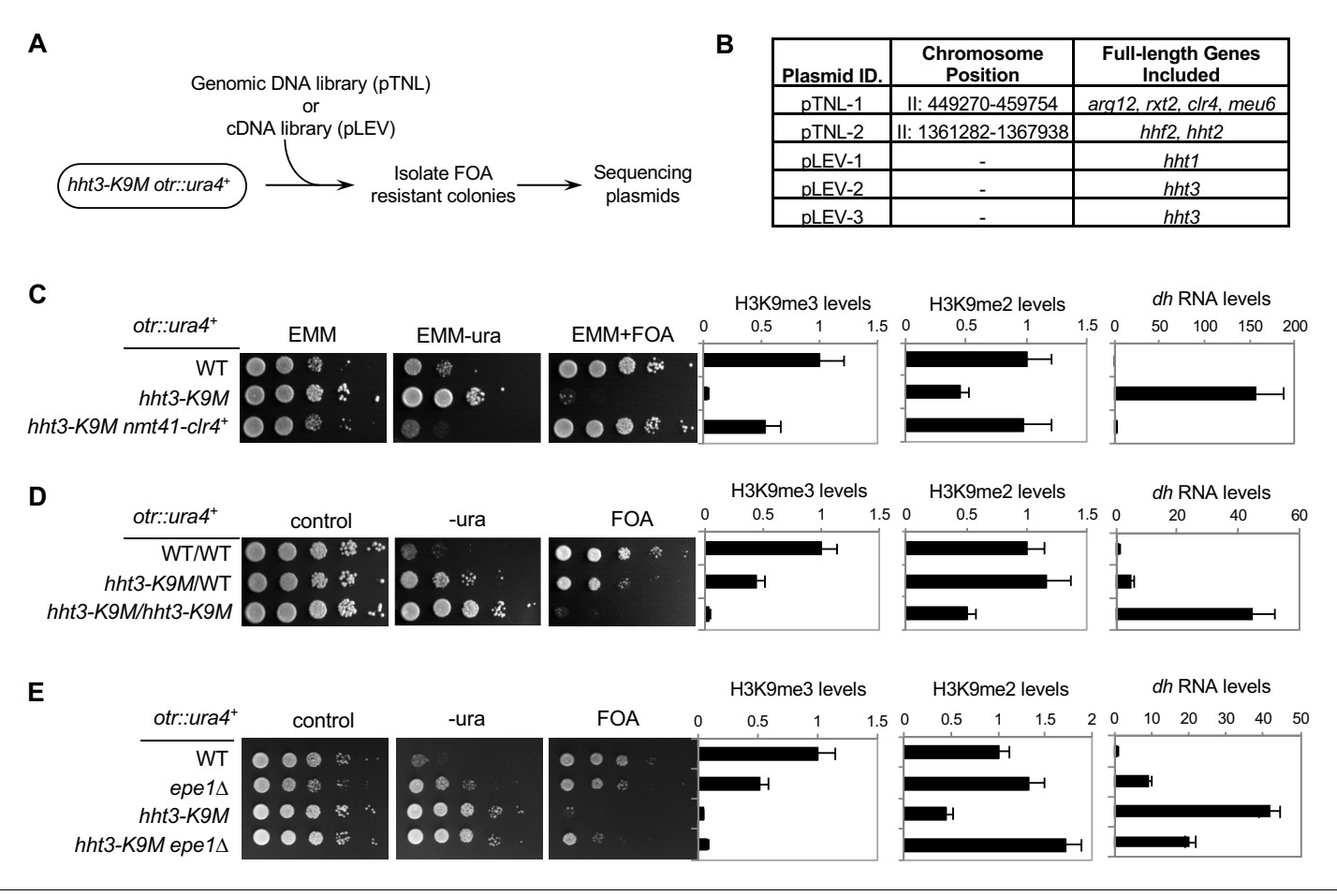

**Figure 6.** The effects of H3K9M on heterochromatin depend on the relative concentrations of H3K9M and Clr4. (**A**) A schematic diagram of the screen procedure for overexpression suppressors of *hht3-K9M*. (**B**) List of genomic and cDNA clones identified in the screen. (**C, D, E**) Left, ten-fold serial dilution analyses of indicated yeast strains grown on indicated media to measure the expression of *otr::ura4⁺*. Right, ChIP analyses of H3K9me3 and H3K9me2 levels at pericentric *dh* repeat, normalized to *act1⁺*, and qRT-PCR analysis of *dh* repeat transcript levels, normalized to *act1⁺*. Data presented is mean ± s.d. of four technical replicates.

The following figure supplement is available for figure 6:

**Figure supplement 1.** Ten-fold serial dilution analyses of indicated yeast strains grown on indicated media to measure the expression of *otr::ura4⁺*.

## Discussions

Histone K-to-M mutations are associated with distinct types of cancers and they function in a dominant fashion to block the methylation of lysine residues. However, the molecular mechanism by which these mutations block histone methylation is not well understood. Mammalian cells contain large number of histone genes (and their variants) and histone methyltransferases, which complicate mechanistic analysis. Here, we established an H3K9M model in a much simpler organism, fission yeast, which contains only three histone H3 genes and a single H3K9 methyltransferase Clr4. We found that cells containing one copy of mutant H3K9M completely abolished H3K9me3 on the two other wild copies of histone H3 across the entire genome.

The formation of heterochromatin is divided into two distinct steps: initiation and spreading. The histone methyltransferase Clr4 is first targeted to heterochromatin nucleation centers to initiate H3K9 methylation. The spreading of heterochromatin requires Clr4 mediated H3K9 methylation, which creates binding sites for the chromo domain proteins such as Swi6 and Clr4, thus enhancing the binding of Clr4 to methylate adjacent histones (*Figure 7*). We found that Clr4 is trapped at

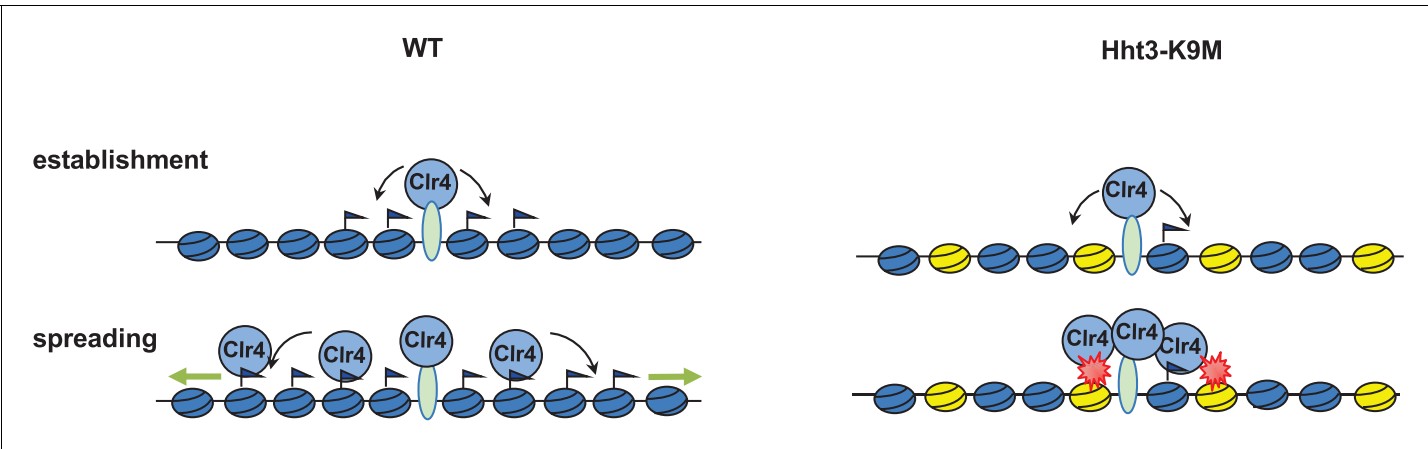

**Figure 7.** A model of H3K9M dominantly blocking histone H3K9 methylation. In wild type cells, Clr4 is first recruited to heterochromatin nucleation centers to initiate H3K9 methylation. Methylated nucleosomes recruit additional Clr4 to methylate neighboring nucleosomes, leading to heterochromatin spreading. In the end, low levels of Clr4 will cover the entire heterochromatin domain. In H3K9M expressing cells, Clr4 is still recruited to heterochromatin nucleation centers, but will be trapped there due to its interaction with H3K9M containing nucleosomes (yellow). This result in the failure of Clr4 spreading to form heterochromatin domains.

heterochromatin nucleation centers in cells containing H3K9M. The fact that H3K9M enhances the interaction between the SET domain of Clr4 and H3 tail, and that compromising such interaction alleviates the trapping of Clr4 by H3K9M to allow heterochromatin assembly suggest that the trapping of Clr4 is mediated by a direct interaction between H3K9M and the Clr4 SET domain. Since Clr4 enzymatic activity is required for heterochromatin spreading and H3K9M inhibits Clr4 activity, it is also possible that the lack of H3K9me also contributes to the failure of Clr4 spreading in addition to the interaction between Clr4 and H3K9M. The fact that trapping only occurs at heterochromatin nucleation centers suggests that recruitment of Clr4 to H3K9M-containing nucleosomes is essential for trapping of Clr4. Alternatively, other factors might also contribute to the trapping of Clr4 in vivo.

It should be noted that in *hht3-K9M* cells, although H3K9me3 is completely abolished from the entire genome, there are still significant amounts of H3K9me2 present at heterochromatin nucleation centers. Given that the interaction between H3K9M and Clr4 is not covalent, Clr4 is expected to be in an equilibrium between H3K9M bound and free states. Therefore, although Clr4 is trapped at heterochromatin nucleation centers for a majority of the time, there will be short intervals when Clr4 is in the free form and thus available to methylate nearby nucleosomes. Since the $k_{cat}$ of Clr4 for H3K9me3 is 10 times slower than that of H3K9me2 (*Al-Sady et al., 2013*), it is much easier for Clr4 to catalyze H3K9me2 than H3K9me3, therefore resulting in a much stronger effects on H3K9me3 than H3K9me2 at heterochromatin nucleation centers, where Clr4 levels are high.

Our structure of G9a in complex with H3K9M and SAM provides the molecular details of how methionine interacts with histone methyltransferases. In contrast to Clr4, which can perform H3K9me3 and regulates heterochromatin assembly, G9a only performs H3K9me2 and mainly acts at euchromatin. Nonetheless G9a is inhibited by H3K9M in vitro (*Lewis et al., 2013*) and its SET domain is highly conserved with that of Clr4. Therefore, we used the G9a-H3K9M-SAM structure as a guide for further analyses. The crystal structure of G9a-H3K9M-SAM complex clearly shows the methionine side chain of H3K9M. It occupies a similar position as that of a normal lysine and interacts with several aromatic residues of G9a through van der Waals interactions. The most obvious difference between the methionine and lysine side chains is the bending of the end methyl group of methionine. When the methionine was substituted with Norleucine, a similar bend of the methyl group was also observed. Such bending increases the diameter of the side chain to enhance van der Waals interactions, therefore explaining the increased binding of H3K9M and H3K9Nle to H3K9 methyltransferases. Although SAM enhanced the binding between H3K9M and H3K9 methyltransferases, SAM does not make strong direct contributions to the interactions with the methionine side chain, suggesting that its role is to stabilize the structure of the protein to properly form the binding

pocket. In addition, the position of SAM is incompatible with dimethylated lysine within G9a, suggesting that the binding of SAM helps release the wild type histone H3 tail after methylation reactions. Therefore, the structure explains why SAM is critical for stabilizing the interaction between H3K9M and H3K9 methyltransferases.

The binding mode of H3K9M-G9a-SAM differs from the structure of H3K27M in complex with *Chaetomium thermophilum* PRC2, in which an adjacent arginine occupies the lysine channel (*Jiao and Liu 2015*), even though H3K9M also has an adjacent arginine. It is consistent with the structure of the H3K27M-human PRC2 complex, with methionine occupying the expected lysine-binding channel (*Justin et al., 2016*). Such a binding state can explain the fact that all known K-to-M mutations, even those without adjacent arginine (such as H3K36M and H3K4M), inhibit the enzymatic activity of their corresponding histone methyltransferases (*Chan et al., 2013b*; *Lewis et al., 2013*; *Herz et al., 2014*). The high conservation of aromatic residues mediating interactions with the methionine side chain suggests that other K-to-M mutations might function in a similar fashion. Indeed, mutant histones show increased association with histone methyltransferases in vivo and in vitro (*Bender et al., 2013*; *Chan et al., 2013a*; *Justin et al., 2016*; *Lu et al., 2016*; *Fang et al., 2016*). The strengthened interaction might similarly prevent the turnover of these enzymes to block the methylation of other wild type histones.

Finally, our results also suggest several ways to detoxify the effects of H3K9M. First, the effect of H3K9M can be modulated by a modification on the K9M containing histone tail that affects its interaction with Clr4. It is interesting to note that H3K27 is also adjacent to a serine (S28). S28 phosphorylation reduced the inhibitory effects of H3K27M on PRC2 activity in vitro, and introduction of a H3K27MS28E mutant reduced the effects of H3K27M on H3K27me3 in vivo (*Brown et al., 2014*). Another way to modulate the effects of H3K9M is through changing Clr4 levels or mutant-to-wild type histone ratios, suggesting that the effects of H3K9M are exerted through a direct competition between H3K9M and normal H3 for a limited pool of Clr4 protein. Finally, tipping the balance between heterochromatin assembly and disassembly, such as the deletion of putative histone demethylase Epe1, can also partially detoxify H3K9M. This is consistent with an earlier observation that inhibiting H3K27 demethylase JMJD3 alleviates the effects of H3K27M mutation in glioblastoma (*Hashizume et al., 2014*). Given the similarities between different histone K-to-M mutations, it is highly likely that these approaches can be used to detoxify other histone K-to-M mutations as well.

## Materials and methods

### Fission yeast strains and genetic analyses

Yeast strains containing epitope tagged histones and their mutants were generated by a PCR-based module method. All other strains were constructed through genetic crosses. A list of yeast strains used is provided in *Supplementary file 2*. For serial dilution plating assays, ten-fold dilutions of a mid log-phase culture were plated on the indicated medium and grown for 3 days at 30°C.

### Chromatin immunoprecipitation (ChIP) analyses

Log-phase yeast cells were incubated at 18°C for 2 hours and then fixed for 30 minutes in 3% freshly made formaldehyde. The crosslinking reaction was stopped by the addition of 2.5 M Glycine to make a final concentration of 0.125 M. The cells were pelleted and washed with PBS (phosphate buffered saline) before resuspended in ChIP lysis buffer (50 mM HEPES-KOH, pH 7.5, 140 mM NaCl, 1% Triton X-100, 0.1% Deoxycholate supplemented with cOmplete protease inhibitors cocktail (Roche)). Ice cold glass beads were added and the mixtures were vigorously disrupted in a bead-beater. The lysates were collected and subjected to sonication to reduce chromatin size to 500–1000 base pairs. The cleared cell lysates were incubated with antibodies: H3K9me3 (Abcam), H3K9me2 (Abcam), and Flag (Sigma) over night at 4°C. Protein G beads were then added to isolate the antibodies and associated chromatin fragments. The beads were then washed with ChIP lysis buffer twice, ChIP lysis buffer containing 0.5 M NaCl, Wash buffer (10 mM Tris, pH 8.0, 250 mM LiCl, 0.5% NP-40, 0.5% Deoxycholate, 1 mM EDTA), and TE (50 mM Tris pH 8.0, 1 mM EDTA). The bound chromatin fragments were eluted with TES (50 mM Tris pH 8.0, 1 mM EDTA, 1% SDS) and the crosslinking was reversed by incubating at 65°C overnight. The protein DNA mixture were then

subjected to proteinase K treatment and phenol:chloroform extraction before the DNA was precipitated by ethanol.

Quantitative real-time PCR (qPCR) was performed with Maxima SYBR Green qPCR Master Mix (Fermentas) in a StepOne Plus Real Time PCR System (Applied Biosystems). DNA serial dilutions were used as templates to generate a standard curve of amplification for each pair of primers, and the relative concentration of target sequence was calculated accordingly. An *act1* fragment was used as a reference to calculate the enrichment of ChIP over WCE for each target sequence. A list of DNA oligos used is provided in *Supplementary file 3*.

For ChIP-seq, DNA samples were prepared according to TruSeq ChIP sample preparation guide (Illumina) and sequenced on the Illumina HiSeq 2500 system by 100 bp paired-end sequencing. The raw reads were trimmed by Trimmomatic (v0.35) (*Bolger et al., 2014*) to remove potential adapter contamination and regions with bad sequencing qualities. The trimmed reads were aligned to the *S. pombe* reference genome (Ensembl version: ASM294v2.29) by bwa (0.7.12-r1039) (http://bio-bwa.sourceforge.net/). The resulting sam files were further processed by Samtools (v1.2) (*Li et al., 2009*), picard-tools (v2.0.1) (http://broadinstitute.github.io/picard/) and GATK (v3.5) (*McKenna et al., 2010*) for indexing, sorting, removing PCR duplicates, and local-realignment. The per-based mapping depth was calculated by bedtools (v2.25.0) (*Quinlan and Hall, 2010*) and the sliding window plots (window size = 100 bp, step size = 50 bp) were created by in-house Perl and R scripts. We also employed MACS (v1.4.2) (*Zhang et al., 2008b*) to contrast each ChIP sample versus WCE sample for peak calling.

## RNA analyses

Total cellular RNA was isolated from log-phase cells using MasterPure yeast RNA purification kit (Epicentre) according to the manufacturer's protocol. Quantification with real time RT-PCR was performed with Power SYBR Green RNA-to-CT one-step Kit (Applied Biosystems). RNA serial dilutions were used as templates to generate a standard curve of amplification for each pair of primers, and the relative concentration of the target sequence was calculated accordingly. An *act1* fragment served as a reference to normalize the concentration of samples. The concentration of each target gene in wild type was arbitrarily set to 1 and served as reference for other samples.

## In vitro histone methyltransferase and peptide binding assays

Histone peptides (H3, 1–21) with a C-terminal biotinylated lysine were synthesized by Anaspec at 90% purity and confirmed by mass spectrometry analyses. Clr4 chromo domain (1–190) and SET domain (190–490) were cloned into a pGEX vector. Expression plasmid for GST tagged Swi6, Chp2, and Chp1 chromo domains are gifts from Dr. Jun-ichi Nakayama. The GST fusion proteins were purified with Glutathione beads (GE Healthcare) according to the manufacturer's protocol.

To examine the enzymatic activity of Clr4 mutants, histone methyltransferase assays were performed with 0.5 μg of recombinant Clr4 SET domain and 2.6 μg of Hela histones in a histone methyltransferase buffer (50 mM Tris, pH 8.0, 1 mM EDTA, 0.5 mM DTT) supplemented with $^3$H-SAM for 30 minutes at 30°C. The samples were resolved by SDS-PAGE and subjected to Coomassie staining to visualize the proteins and then treated with EN3HANCE (Perkin Elmer) to visualize labeled substrates.

To examine the inhibitory effects of H3K9M peptide, 0.1 μg recombinant SET domain of Clr4 was incubated with 1 μg recombinant human nucleosomes in a methyltransferase buffer (100 mM Tris, pH 8.8, 100 mM KCl, 5% glycerol, 1 mM MgCl$_2$, 20 μM ZnSO$_4$, 10 mM β-mercaptoethanol) supplemented with 10 μM non-radioactive SAM and 2 μM [$^3$H]-labeled SAM for 30 minutes at 30°C. The H3K9M peptide was incubated with Clr4 SET domain for 15 minutes at 30°C before the addition of mono-nucleosomes.

Protein binding assays with recombinant chromo domains were performed by incubating recombinant proteins with biotinylated histone peptides in PBS buffer for 30 minutes at 30°C. Streptavidin beads were added to isolate biotinylated peptide and associated proteins. The beads were washed three times with PBS buffer. The proteins bound to the beads were resolved by SDS-PAGE and stained with Coomassie blue.

Protein binding assays with recombinant SET domains were performed by incubating recombinant proteins in histone methyltransferase buffer (50 mM Tris, pH 8.0, 50 mM NaCl, 1 mM EDTA,

0.5 mM DTT) supplemented with 50 µM SAM or SAH for one hour at 30°C. Streptavidin beads were added to isolate biotinylated peptide and associated proteins. The beads were washed three times in histone methyltransferase buffer before resolved by SDS-PAGE and stained with Coomassie blue. To examine whether the binding was covalent, ChIP lysis buffer (50 mM HEPES-KOH, pH7.5, 500 mM NaCl, 1% Triton X-100, 0.1% sodium deoxycholate) was used to perform the washes.

## Protein expression and purification

The expression and purification of the SET domain of human G9a (residues 913–1193) followed a published procedure (*Wu et al., 2010*). The expression plasmid (gift of Prof. Matthieu Schapira) was transformed into competent cells and induced using 0.25 mM isopropyl-1-thio-D-galactopyranoside when A600 is 0.7. After incubation overnight at 16°C, the cells were harvested and resuspended in lysis buffer (20 mM Tris (pH 8.0), 250 mM NaCl, and 5% (v/v) glycerol) supplemented with 2 mM β-mercaptoethanol, 0.1% Igepal and 1 mM phenylmethylsulfonyl fluoride and lysed with ultrasonication. The lysate was incubated with Ni-NTA resin (QIAGEN) and washed with 10 column volume lysis buffer containing 50 mM imidazole. Bound protein was eluted with lysis buffer containing 250 mM imidazole, and was digested overnight with thrombin to remove the 6xHis-tag. After concentration, the protein was loaded onto a Superose 6 column (GE Healthcare) equilibrated with gel filtration buffer (20 mM Tris (pH 8.0), 150 mM NaCl). The fractions of the protein peaks were collected and concentrated.

## Crystallization

The protein was incubated with the H3K9M peptide at a molar ration of 1:10 on ice for 30 min, and the sample was then set up for crystallization with the hanging-drop vapor diffusion method at 20°C. The reservoir solution contained 0.1 M Bis-Tris propane (pH 7.5), 18% (w/v) PEG3350, 0.2 M NaF, and 5% (v/v) ethylene glycol. Using mother liquor supplemented with 15% (v/v) glycerol as cryo-protectant, the crystals were frozen in liquid nitrogen before data collection.

## Data collection and structure determination

X-ray diffraction data were collected at 100K at NE-CAT beamline 24-ID-C of Advanced Photon Source (APS) at Argonne National Laboratory and processed using program HKL2000 (*Otwinowski and Minor, 1997*). The structure was solved through molecular replacement using Phaser-MR in program PHENIX (*Adams et al., 2002*). The structure refinement was carried out using PHENIX, and manual model building with Coot (*Emsley and Cowtan, 2004*).

## Accession numbers

ChIP-seq data is available at Arrayexpress under accession number E-MTAB-4776. The atomic coordinates have been deposited at the Protein Data Bank, with accession codes 5T0K (Clr4-H3K9M-SAM) and 5T0M (Clr4-H3K9Nle-SAM).

## Acknowledgements

We thank Jun-ichi Nakayama for providing chromo domain expression plasmids, Matthieu Schapira for G9a plasmid, National BioResource Project (Japan) for providing the genomic DNA library, Charlie Hoffman for the cDNA library, and members of Jia lab for helpful discussions and critical reading of the manuscript. This work was supported by National Institutes of Health grants R01-GM085145 to SJ, S10-OD012018 to LT, the National Center for Research Resources (P41-RR011823), and National Institute of General Medical Sciences (P41-GM103533). J-XY is supported by a postdoctoral fellowship from Fondation ARC pour la Recherche sur le Cancer (n°PDF20150602803). This work is based upon research conducted at the Northeastern Collaborative Access Team beamlines, which are funded by the National Institute of General Medical Sciences from the National Institutes of Health (P41 GM103403). The Pilatus 6M detector on 24-ID-C beam line is funded by a NIH-ORIP HEI grant (S10 RR029205). This research used resources of the Advanced Photon Source, a U.S. Department of Energy (DOE) Office of Science User Facility operated for the DOE Office of Science by Argonne National Laboratory under Contract No. DE-AC02-06CH11357.

# Additional information

## Funding

| Funder | Grant reference number | Author |
|---|---|---|
| National Institutes of Health | R01-GM085145 | Songtao Jia |
| National Institutes of Health | S10-OD012018 | Liang Tong |
| National Institutes of Health | P41-RR011823 | John R Yates III |
| National Institutes of Health | P41-GM103533 | John R Yates III |
| Fondation ARC pour la Recherche sur le Cancer | PDF20150602803 | Jia-Xing Yue |

The funders had no role in study design, data collection and interpretation, or the decision to submit the work for publication.

## Author contributions

C-MS, JW, Designed experiments, Performed experiments, Analyzed data; KX, Performed crystallography analyses; HC, Performed ChIP-seq; J-XY, SA, PLN, Performed ChIP-seq data analyses; JJM, JRY, Performed mass spectrometry analyses; LT, SJ, Designed experiments, Analyzed data, Wrote the paper

## Author ORCIDs

Jia-Xing Yue, http://orcid.org/0000-0002-2122-9221
Songtao Jia, http://orcid.org/0000-0002-7927-0227

# Additional files

## Supplementary files

• Supplementary file 1. Data collection and refinement statistics.

• Supplementary file 2. Yeast strains used in this study.

• Supplementary file 3. DNA oligos used in this study.

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
