## [Decision Letter]

Thank you for submitting your article "A histone H3K9M mutation traps histone methyltransferase Clr4 to prevent heterochromatin spreading" for consideration by *eLife*. Your article has been reviewed by three peer reviewers, one of whom, Jerry Workman, is a member of our Board of Reviewing Editors, and the evaluation has been overseen Jessica Tyler as the Senior Editor.

The reviewers have discussed the reviews with one another and the Reviewing Editor has drafted this decision to help you prepare a revised submission.

This is a timely manuscript that seeks to explain how an H3K9M mutation in a single copy of histone genes hinders methylation of WT H3 in the same cell. The data support a model whereby the H3K9M mutation traps the Clr4 methyltransferase preventing its spreading to form heterochromatic domains. This is a novel mechanism based on genetics in *S. pombe* and crystallography.

Summary:

Recently, histone H3.3K27M and H3.3K36M mutations have been found in high-grade pediatric brain tumors and chondroblastoma of bone, respectively. In addition, it has been shown that expression of histone K to M mutant transgenes in human cells leads to the reduction in methylation of corresponding lysine residues on endogenous histones. However, the mechanism of action of histone K to M mutant remains in debate in the field. The manuscript by Shan et al. utilizes fission yeast as the model system to address the mechanism whereby H3K9M affects H3K9 methylation. Expression of H3K9M from one of the three histone H3 genes results in complete loss of H3K9me3 and silencing at heterochromatin. Interestingly, Clr4, the only enzyme for H3K9 methylation remains bound to nucleation center, where H3K9me2, but not H3K9me3, was detected. In vitro, Clr4 and G9a bound to H3K9M peptides with higher affinity than the corresponding wild type H3. They also solved the crystal structure of G9a catalytic domain in complex with H3K9M peptide. Based on the structure, they tested several possible ways to relieve the inhibition of Clr4 activity by H3K9M proteins.

The authors build a model in which the block in heterochromatin assembly is mediated solely via sequestration of Clr4 by its SET domain binding to H3K9M. I am not totally convinced that this is the case, and would like to see further work done to rule out contribution of Clr4 CD in K9M mediated disruption of silencing.

Finally, G9a structure with K9M peptide was recently published (Jayaram et al. 2016 PNAS) including demonstration of importance of SAM for conformational change, which reduces impact of some of this story.

Overall, this manuscript is well written, interesting, timely and will help resolve the debate in the field. We have the following suggestions to improve the manuscript.

Essential revisions:

1) It is a bit disconcerting that when H3K9M is supposed to inhibit Clr4 activity in vivo, that such inhibition could not be observed in vitro. While it is always possible that it won't work in vitro, it does suggest that other components in vivo are involved reducing enthusiasm for the author’s simple model. Perhaps more effort should be devoted to optimizing the biochemical assay. It is stated that H3K9M peptide cannot inhibit enzymatic activity of Clr4 in vitro when peptides and octamers were used as substrates. Did the authors use mononucleosomes as substrates?

2) Another point regarding the in vitro methylation assays. Was recombinant full length Clr4 tried? (it has HMT activity: Nakayama et al. Sci 2001). I have a nagging feeling that Clr4's CD could be critical to the suppression of Clr4 activity by K9M. It is known that the detailed mechanism of action of Clr4 differs from Suv39h1 (compare Al-Sady et al. Mol Cell 2013 with Muller et al., Nat Chem Biol 2016) and so presumably also G9a (which only performs dimethylation and not trimethylation). As the chromodomain is also suggested to be involved in heterochromatin spreading it is important that the authors determine the extent that loss of chromodomain interaction contributes to the H3K9M phenotypes.

3) Something else to consider is that *pombe* has 4 CD proteins that can bind H3K9 when methylated. The authors perform binding studies with Clr4 CD- and show as previously reported that it has higher affinity for K9me3 than me2 peptides. They see no interaction with H3K9M. But there are some inconsistencies with description of this data such as they say that SET domain interactions with H3K9M (which they can detect) are not covalent as they are disrupted by 500 mM Nacl wash with detergents, but list this same buffer as their wash buffer for the pull down experiments.

Do other CD proteins bind K9M and does this contribute to the heterochromatin defects in K9M expressing cells?

4) Why is K9me2 detected at "nucleation" points, but not K9me3? If the model is that K9M binds and blocks Clr4's SET domain, wouldn't the expected result be equivalent loss of all forms of methylated H3? Does this just then reflect a difference in the normal levels of the different methylated forms of H3 in cells or reflect a difference in the sensitivity of detection of the different methylated forms in the ChIP assays? Or is it telling us something about the mechanism of inhibition of HMT activity. I think this is a case where quantitative spike in of dNucs could be useful- or at least referencing of others work in this area (Al-Sady et al. Mol Cell 2013) who show that the dimethylated K9 state is predominant in *pombe*.

5) The only piece of evidence presented that the inhibition is mediated via the SET domain of Clr4 is disruption of silencing of a GAL4-reporter in K9M expressing cells that have GBD-Clr4 tethered (that lacks the CD of Clr4). Very little data is shown for this (the system shows variegation in WT cells, and it is difficult to assess what is going on in the population when the color of single colonies are shown). Fields of colonies should be presented. Similarly, were "white" and "red" colonies picked for the ChIP analysis or was this from a mixed population?

I think a nice experiment that would help cement that the K9M effect is via Clr4 SET domain would be demonstration that this GBD-Clr4 fusion is recruited to nucleation sites at sites of heterochromatin in cells lacking the GBD reporter. This would reinforce that the mechanism relies on SET domain interaction with K9M.

Alternately, there are 2 CD mutant versions of Clr4 (in Nakayama et al. 2001) that in WT cells retain HMT activity at centromeres. These are aromatic cage mutants of the CD- so if K9M is binding via the CD – these mutants may be insensitive to K9M-mediated disruption of centromeric silencing.

---

## [Author Response]

*Summary:*

*Recently, histone H3.3K27M and H3.3K36M mutations have been found in high-grade pediatric brain tumors and chondroblastoma of bone, respectively*

[…]

*The authors build a model in which the block in heterochromatin assembly is mediated solely via sequestration of Clr4 by its SET domain binding to H3K9M. I am not totally convinced that this is the case, and would like to see further work done to rule out contribution of Clr4 CD in K9M mediated disruption of silencing.*

We thank the reviewers for their constructive suggestions. In the revised manuscript, we have additional evidence to rule out the contribution of Clr4-CD. First, we optimized in vitro histone methyltransferase assays and showed that the Clr4 SET domain is inhibited by H3K9M (Figure 2). Second, the Clr4 chromo domain mutant (W31G) is also trapped by H3K9M at heterochromatin nucleation centers in vivo (Figure 3—figure supplement 2). These new data, together with in vitro binding assays showing that the SET domain, but not the CD, binds to H3K9M, clearly demonstrate that H3K9M function through the SET domain of Clr4.

*Finally, G9a structure with K9M peptide was recently published (Jayaram et al. 2016 PNAS) including demonstration of importance of SAM for conformational change, which reduces impact of some of this story.*

Our work is completely independent of that report (which was contributed to PNAS by an NAS member, submitted for review on March 16 and accepted on April 8. Our manuscript was already under consideration for publication at that time (at a different journal)), and reveals mechanistic insights in a living organism well beyond the structural information, which are not available in that report.

*Overall, this manuscript is well written, interesting, timely and will help resolve the debate in the field. We have the following suggestions to improve the manuscript.*

*Essential revisions:*

*1) It is a bit disconcerting that when H3K9M is supposed to inhibit Clr4 activity* in vivo*, that such inhibition could not be observed* in vitro*. While it is always possible that it won't work* in vitro*, it does suggest that other components* in vivo *are involved reducing enthusiasm for the author’s simple model. Perhaps more effort should be devoted to optimizing the biochemical assay. It is stated that H3K9M peptide cannot inhibit enzymatic activity of Clr4* in vitro *when peptides and octamers were used as substrates. Did the authors use mononucleosomes as substrates?*

We optimized conditions for in vitro histone methyltransferase assays and found that the H3K9M peptide inhibits Clr4 enzymatic activity when recombinant mono-nucleosome was used as substrates. One likely reason for the stronger inhibitory effects on nucleosomes is that the activity of Clr4 is much weaker on nucleosomes compared with histone octamers, thus providing less opportunity for Clr4 to complete methylation reactions when Clr4 is in an H3K9M-free state.

*2) Another point regarding the* in vitro *methylation assays. Was recombinant full length Clr4 tried? (it has HMT activity: Nakayama et al. Sci 2001). I have a nagging feeling that Clr4's CD could be critical to the suppression of Clr4 activity by K9M. It is known that the detailed mechanism of action of Clr4 differs from Suv39h1 (compare Al-Sady et al. Mol Cell 2013 with Muller et al., Nat Chem Biol 2016) and so presumably also G9a (which only performs dimethylation and not trimethylation). As the chromodomain is also suggested to be involved in heterochromatin spreading it is important that the authors determine the extent that loss of chromodomain interaction contributes to the H3K9M phenotypes.*

The new histone methyltransferase assay in Figure 2 shows that the SET domain of Clr4 is inhibited by H3K9M. Furthermore, our binding data in Figure 3 showed that the Clr4-SET domain, but not the Clr4-CD, binds to an H3K9M peptide. These results demonstrate that the effect of H3K9M is through the SET domain. We did not use full length Clr4 protein because it is not expressed well in *E. coli*. Given that our new data has already ruled out the possibility that Clr4-CD contributes to the phenotypes associated with the H3K9M mutation, it is not necessary to further test the full length Clr4.

We also examined the localization of Clr4 chromo domain mutant W31G, as suggested by the reviewer. We found that Flag-Clr4-W31G levels at *cenH* increased in *hht3-K9M* cells (new Figure 3—figure supplement 2). Together with our biochemical analysis showing that H3K9M peptide interacts directly with the SET domain, these data support our main conclusion that H3K9M functions through the SET domain rather than the chromo domain of Clr4.

The difference between Clr4 and SUV39H1 mainly lies at a region at the N-terminus of SUV39H1, which is absent from Clr4. Since we have shown that the SET domain is inhibited by H3K9M in vitro, and that the chromo domain mutant is trapped at heterochromatin nucleation center by H3K9M, such a difference is irrelevant to the study here.

*3) Something else to consider is that pombe has 4 CD proteins that can bind H3K9 when methylated. The authors perform binding studies with Clr4 CD- and show as previously reported that it has higher affinity for K9me3 than me2 peptides. They see no interaction with H3K9M. But there are some inconsistencies with description of this data such as they say that SET domain interactions with H3K9M (which they can detect) are not covalent as they are disrupted by 500 mM Nacl wash with detergents, but list this same buffer as their wash buffer for the pull down experiments.*

*Do other CD proteins bind K9M and does this contribute to the heterochromatin defects in K9M expressing cells?*

The high salt buffer with detergent was only used to test whether the H3K9M-Clr4 interaction was covalent, not in any other binding assays. In the revised manuscript, we listed the buffer used for each binding assay in figure legends to avoid any confusion.

As suggested by the reviewer, we performed binding assays with recombinant chromo domains of the other three chromo domain containing proteins (Swi6, Chp1, and Chp2) involved in heterochromatin formation (Figure 3—figure supplement 1). These proteins all interacted with H3K9me2 and H3K9me3 peptides, but not with H3K9M peptide. The experiments in Figure 6 and Figure 6—figure supplement 1, in which overexpression of Clr4, but not Swi6, could not rescue H3K9M phenotypes, further support the idea that H3K9M functions through Clr4 instead of chromo domain proteins such as Swi6.

*4) Why is K9me2 detected at "nucleation" points, but not K9me3? If the model is that K9M binds and blocks Clr4's SET domain, wouldn't the expected result be equivalent loss of all forms of methylated H3? Does this just then reflect a difference in the normal levels of the different methylated forms of H3 in cells or reflect a difference in the sensitivity of detection of the different methylated forms in the ChIP assays? Or is it telling us something about the mechanism of inhibition of HMT activity. I think this is a case where quantitative spike in of dNucs could be useful- or at least referencing of others work in this area (Al-Sady et al. Mol Cell 2013) who show that the dimethylated K9 state is predominant in pombe.*

We have added a paragraph in the Discussion section (third paragraph) regarding the different effects of H3K9M on H3K9me3 and H3K9me2. Our structural and biochemical analyses indicate that H3K9M enhances binding of histone H3 to Clr4, but the binding is not covalent. Therefore, Clr4 should be in an equilibrium between H3K9M-bound and free states. Al-Sady et al., 2013 have beautifully demonstrated that the *k_cat_*of Clr4 to catalyze H3K9me3 is 10 times slower than that of H3K9me2, resulting in much higher in vivo H3K9me2 levels than H3K9me3. Due to the slow rate of H3K9me3, the time interval that Clr4 falls off H3K9M might only allow it to perform H3K9me2, but difficult to complete H3K9me3.

Spike in of dNucleosomes is used to calculate the absolute levels of H3K9me3 and H3K9me2 in cells. Given that we are only comparing the relative levels of H3K9me3 and H3K9me2 between wild type and *hht3-K9M* cells, we believe that the experiments with dNucleosomes are unnecessary. We referenced Al-Sady et al. as suggested by the reviewer.

*5) The only piece of evidence presented that the inhibition is mediated via the SET domain of Clr4 is disruption of silencing of a GAL4-reporter in K9M expressing cells that have GBD-Clr4 tethered (that lacks the CD of Clr4). Very little data is shown for this (the system shows variegation in WT cells, and it is difficult to assess what is going on in the population when the color of single colonies are shown). Fields of colonies should be presented. Similarly, were "white" and "red" colonies picked for the ChIP analysis or was this from a mixed population?*

The system we used only showed very little variation. We included pictures of fields of colonies in Figure 2—figure supplement 1). In the absence of GBD-Clr4-∆CD, all colonies are white. In the presence of GBD-Clr4-∆CD, the vast majority of colonies are red, with only a very small number of white colonies. In the presence of GBD-Clr4-∆CD and Hht3-K9M, all colonies are white.

Our ChIP analyses were performed with mixed population of cells without any selection.

*I think a nice experiment that would help cement that the K9M effect is via Clr4 SET domain would be demonstration that this GBD-Clr4 fusion is recruited to nucleation sites at sites of heterochromatin in cells lacking the GBD reporter. This would reinforce that the mechanism relies on SET domain interaction with K9M.*

*Alternately, there are 2 CD mutant versions of Clr4 (in Nakayama et al. 2001) that in WT cells retain HMT activity at centromeres. These are aromatic cage mutants of the CD- so if K9M is binding via the CD – these mutants may be insensitive to K9M-mediated disruption of centromeric silencing*

As suggested by the reviewer, we examined the chromo domain mutant W31G of Clr4. We found that Flag-Clr4-W31G levels at the silent mating-type region heterochromatin nucleation center *cenH* increased in *hht3-K9M* cells (new Figure 2—figure supplement 3). This result further supports our main conclusion that H3K9M functions through the SET domain rather than the chromo domain of Clr4.